# The Anti-Inflammatory Effects of *Cannabis sativa* Extracts on LPS-Induced Cytokines Release in Human Macrophages

**DOI:** 10.3390/molecules28134991

**Published:** 2023-06-25

**Authors:** Mariia Zaiachuk, Santosh V. Suryavanshi, Nazar Pryimak, Igor Kovalchuk, Olga Kovalchuk

**Affiliations:** Department of Biological Sciences, University of Lethbridge, Lethbridge, AB T1K 3M4, Canada

**Keywords:** cannabidiol, cannabis, inflammation, lipopolysaccharide, macrophages, tetrahydrocannabinol

## Abstract

Inflammation is the response of the innate immune system to any type of injury. Although acute inflammation is critical for survival, dysregulation of the innate immune response leads to chronic inflammation. Many synthetic anti-inflammatory drugs have side effects, and thus, natural anti-inflammatory compounds are still needed. *Cannabis sativa* L. may provide a good source of anti-inflammatory molecules. Here, we tested the anti-inflammatory properties of cannabis extracts and pure cannabinoids in lipopolysaccharide (LPS)-induced inflammation in human THP-1 macrophages. We found that pre-treatment with cannabidiol (CBD), delta-9-tetrahydrocannabinol (THC), or extracts containing high levels of CBD or THC reduced the level of induction of various cytokines. The CBD was more efficient than THC, and the extracts were more efficient than pure cannabinoids. Finally, IL-6, IL-10, and MCP-1 cytokines were most sensitive to pre-treatments with CBD and THC, while IL-1β, IL-8, and TNF-α were less responsive. Thus, our work demonstrates the potential of the use of cannabinoids or/and cannabis extracts for the reduction of inflammation and establishes IL-6 and MCP-1 as the sensitive markers for the analysis of the effect of cannabinoids on inflammation in macrophages.

## 1. Introduction

The positive impact of cannabis on human health has been known for many years [1]. The *C. sativa* plant contains many metabolites, but the most well-known are cannabinoids, terpenes, and flavonoids. Cannabinoids act via the endocannabinoid system (ECS), which consists of receptors such as cannabinoid 1 (CB1) and cannabinoid 2 (CB2), transient receptor potential channels of the vanilloid subtypes 1 and 2 (TRPV1, TRPV2), G protein-coupled receptors 18, 55, and 119 (GPR18, GPR55, GPR119), endocannabinoids such as 2-arachidonoylglycerol and anandamide (2-AG, AEA), and their biosynthesis and degradation enzymes. Homeostasis maintenance is the main function of ECS [2]. The most studied phytocannabinoids produced by *C. sativa* plants are cannabidiol (CBD) and Δ9-tetrahydrocannabinol (THC). The effect of cannabis extracts is frequently different from that of single cannabinoids, with extracts often being more efficient due to the complex interaction between minor cannabinoids, flavonoids, and terpenes—this is known as an entourage effect [3]. The main cannabinoid receptors, CB1 and CB2, are highly expressed in immune cells such as macrophages, T lymphocytes, and B cells, as well as in the cells of the lungs and the bronchial tissues [4,5]. Therefore, cannabis extracts, as well as their active ingredients, can modulate the function of the immune system and inflammation in general.

Inflammation is the body’s vital defense mechanism. Epithelial cells and macrophages are often the first line of defense against bacteria, viruses, and toxins [6]. Bacteria express pathogen-associated molecular patterns (PAMPs) that are recognized by pathogen recognition receptors (PRRs) present on immune and non-immune cells. Lipopolysaccharide (LPS), being a component of the gram-negative bacterial cell wall, is an example of a PAMP [7]. Chronic inflammation plays an important role in the pathogenesis of many chronic conditions, such as inflammatory bowel diseases, obesity, diabetes mellitus, atherosclerosis, asthma, Alzheimer’s disease, and many others [8]. The uncontrolled and continuous release of pro-inflammatory cytokines is associated with organ damage and chronic disease progression [9]. The most frequently induced cytokines are TNF-α, IL-1β, IL-1, IL-6, IL-8, IL-10, monocyte chemoattractant protein-1 (MCP-1), and IL-8. The pro-inflammatory cytokines (for instance, MCP-1 and IL-8) induce chemotaxis of neutrophils, monocytes, T cells, and dendritic cells to the site of infection and stimulate phagocytosis [10,11]. TNF-α and IL-1β are key mediators of the inflammatory response [12,13]. IL-6 stimulates inflammation and induces the synthesis of acute-phase proteins [14]. Anti-inflammatory cytokines, such as IL-10, function to limit the host immune response to pathogens, maintain tissue homeostasis, and prevent the development of autoimmune conditions [15]. The balance between pro-inflammatory and anti-inflammatory cytokines is crucial for proper tissue homeostasis. 

The anti-inflammatory effects of cannabis extracts, as well as their active ingredients, are well documented [16]. For instance, cannabis users suffering from multiple sclerosis had lower levels of pro-inflammatory cytokines (TNF-α, IL-1, IL-6, and IFN-γ) and higher levels of anti-inflammatory cytokines (IL-10) as compared with non-cannabis users [17]. Cannabis extracts high in CBD and THC suppressed the expression of IL-6, IL-1β, MCP-1, and TNF-α genes induced by TNF-α/IFN-γ in human 3D EpiDermFT tissue [18]. Cannabis extract high in CBD was demonstrated to be more potent than pure CBD in suppressing inflammation by preventing TNF-α production and reducing pain and paw swelling in the zymosan-induced inflammation mouse model [19]. Another study demonstrated that high-CBD cannabis extract was superior to CBD in reducing IL-8 and IL-6 levels in vitro in the A549 alveolar epithelial cell line [20]. In addition, cannabis extracts high in CBD and THC were shown to be more effective than pure CBD and THC in graft versus host disease in an allogeneic bone marrow transplantation model by reducing inflammation and improving survival rates [21]. Our previous work also demonstrated that cannabis extracts were efficient in reducing inflammation in skin fibroblasts, lung fibroblasts, and human 3D tissues [18,22,23].

In this work, we attempted to study the effects of cannabis extracts on LPS-induced cytokine release in human THP-1 leukemia monocytes. We aimed to establish a model of the induction of inflammation and hypothesized that cannabinoids would prevent a strong induction of inflammation. We further hypothesized that extracts differ from pure cannabinoids in their anti-inflammatory properties. Here, we found that inflammation was induced efficiently at 3 h after application of LPS and was partially prevented by exposure to CBD and various cannabis extracts. We also noted that IL-6, IL-10, and MCP-1 were more sensitive to pre-treatments with cannabinoids as compared with IL-1β, IL-6, and TNF-α.

## 2. Results

### 2.1. Analysis of Cannabinoids Content Using HPLC 

We first tested the CBD and THC concentrations of three extracts we planned to use in this study. We found that extracts #14 and #178 were high in THC, while #131 was a high CBD extract (Table 1). We also calculated the amount of CBD and THC (in µM) in the extracts at a concentration of 7 µg/mL. This concentration was found to not be toxic to normal skin and lung fibroblasts [23].

We hypothesized that terpene composition and concentration may affect anti-inflammatory properties and thus tested the terpene profile (Table 2). Cultivar #14 was dominant in trans-caryophyllene and limonene; cultivar #131 was dominant in δ-3-carene, α-pinene, β-pinene, and limonene; and cultivar #178 was high in α-pinene, β-pinene, β-myrcene, and α-bisabolol.

### 2.2. Analysis of Time of Induction of Inflammation by LPS

To establish the inflammation model, we tested the time of induction of pro-inflammatory cytokines. We exposed human THP-1 macrophages to LPS for 3 and 6 h; we used 0.5 μg/mL of LPS as this concentration was shown to be effective in inducing inflammation in cell models [24]. We then analyzed the expression of NFκB and p-NFκB using Western blotting. Our aim was to observe upregulation in p-NFκB expression and the corresponding downregulation of NFκB expression to assume that our in vitro inflammation model is working. The quantification of the p-NFκB/NFκB ratio showed that the desired inflammatory response was attained at 3 h post-LPS stimulation (Figure 1). 

After determining that 3 h is the best time for the induction of inflammation by the LPS, we attempted to find out which sample was better for the analysis: the cell culture supernatant (media) or the cell lysate. We exposed cells to LPS for 3 h, harvested media and cell lysates separately, and performed a multiplex ELISA experiment. We found that LPS media samples (3 h of treatment) had much higher levels of IL-1β, TNF-α, and IL-10 cytokines in comparison with LPS cell lysate samples (Appendix A, Appendix A). It was concluded that media samples are better for cytokine analysis, which is why we used cell media for Multiplex ELISA in the analysis of the effect of cannabinoids on inflammation. 

### 2.3. Treatment with Selected Concentrations of LPS, THC, CBD, and Extracts Does Not Reduce the Viability of Macrophages

To test the effect of cannabis extracts and pure cannabinoids, CBD and THC, on the reduction of inflammation triggered by LPS, we first tested the viability of THP-1 macrophages in response to LPS (0.5 µg/mL), CBD (5 µM), THC (5 µM), or the aforementioned extracts (7 μg/mL). We found that the viability of THP-1 macrophages was not changed in response to any treatment (Appendix A, Appendix A). The cell morphology also did not change. This is important because the level of inflammation can change due to the impact of active ingredients on cell viability. 

### 2.4. Cannabis sativa Extracts Attenuate TNF-α, IL-1β, IL-8, MCP-1, IL-10, and IL-6 Production in LPS-Stimulated THP-1 Macrophages

As explained in the methods, after differentiation, macrophages were pre-treated with cannabinoids or cannabis extracts, and LPS was added 30 min later. Cell supernatants were evaluated for levels of cytokines 3 h after LPS exposure.

LPS treatment drastically increased the levels of all cytokines. CBD pre-treatment significantly decreased the levels of IL-1β, while high CBD extract #131 did not (Figure 2A). Pre-treatment with THC decreased IL-1β, but the change was not significant, while high-THC-containing extracts #14 and #178 decreased it significantly (Figure 2B). The overall increase in IL-6 was lower compared with other cytokines in all tested groups. Although both pure cannabinoids and cannabis extracts significantly reduced IL-6 levels, the cannabis extracts were more efficient (Figure 2C,D). Pure THC did not affect the levels of IL-8 post-LPS; however, pure CBD, high CBD, and high THC extracts significantly reduced IL-8 levels. (Figure 2E,F). Curiously, pure CBD or THC did not prevent the increase in MCP-1, while extracts have done this very effectively (Figure 2G,H). The increase in TNF-α was significantly reduced by CBD but not by THC, while extracts again were more efficient (Figure 2I,J). CBD prevented the increase in IL-10 while THC did not, and again, the extracts have done this more efficiently. 

We summarized our findings in Table 3 below. It appears that CBD is more efficient than THC at preventing the increase in cytokine release. Moreover, it is apparent that the extracts are more efficient than CBD or THC, and this was especially apparent in the THC group comparison. 

We found that high THC extracts (#14 and #178) were more efficient than high CBD extract #131 in reducing the levels of IL-1β, IL-6, and TNF-α after LPS stimulation (Figure 2). In contrast, high CBD extracts were better than high THC extracts at preventing the increase in IL-10 levels (Figure 2). The responses of IL-8 and MCP-1 were comparable in all extracts. 

## 3. Materials and Methods

### 3.1. Main Reagents

The Δ9-THC (Cat#T4764) and CBD (Cat#C-045) were purchased from Sigma-Aldrich (Oakville, ON, Canada). The 1 mg/mL stock solutions of cannabinoids were obtained by dissolving them in methanol and then keeping them at −20 °C.

Lipopolysaccharide (Cat#L4391) was purchased from Sigma. The sterile PBS was used for dissolving LPS to obtain 1 mg/mL stock solution. Phorbol-12-myristate-13-acetate (PMA) was purchased from Sigma-Aldrich (Cat#524400, Oakville, ON, Canada). DMSO (Dimethyl sulfoxide anhydrous, Life Technologies) was used to obtain 1 mg/mL stock solution. Then, the sterile PBS was used to obtain 5 μg/mL stock solution. Trypan Blue Solution (Cat#15250061), 0.4%, was purchased from Thermo Fisher Scientific (Waltham, MA, USA).

### 3.2. Plant Growth and Extract Preparation

All cannabis plants were grown at the University of Lethbridge in the licensed facility. Cannabis sativa cultivars #14, #131, and #178 were used for the experiments. Plants were grown at 22 °C, 18 h light, 6 h dark, for a period of 4 weeks, with 4 plants per cultivar. To induce flowering, plants were shifted to chambers with 12 h light/12 h dark regime. Then the flowers were harvested and dried. Flowers from four plants were combined and then used for the extraction. Three grams of powdered plant tissue were used for the extraction of each studied cultivar. The prepared material was placed into a 250 mL Erlenmeyer flask, and then 100 mL of ethyl acetate was added to each flask. Next, flasks covered in tin foil were incubated overnight at 21 °C on a shaker at 120 rpm. The extracts were filtered and concentrated with the help of a rotary vacuum evaporator and then transferred to a tared 3-dram vial. The solvent was eliminated by evaporating to dryness in an oven overnight at 50 °C. The DMSO was used for the dissolution of crude extracts to obtain 60 mg/mL concentration, and extracts were kept at −20 °C. The complete culture medium was used to obtain the desired concentration of extracts (7 μg/mL). Before use, the extracts were filtered with 0.22 μm filter [18].

### 3.3. High Performance Liquid Chromatography (HPLC) Analysis of Cannabinoids 

Agilent Technologies 1200 Series HPLC system was used for analyzing the levels of THC and CBD in each extract. Software ChemStation LC 3D Rev B.04.02 (Agilent Technologies, Santa Clara, CA, USA) was used for the data acquisition, control of the instrument, and integration. A 9:1 methanol/chloroform (% *v/v*) ratio was used as the mobile phase of the HPLC. Calibration of standards and sample analysis were conducted using 2 μL of the injection volume. The detection of compound peaks was performed at 230 nm and 280 nm. For each cannabis cultivar, two samples with two technical replicates per treatment were analyzed [18]. LoQ (wt%): THC—0.073, THCA—0.06, CBD—0.065, and CBDA—0.1.

### 3.4. Terpene Analysis

We used 2 g of flower samples. Terpene analysis was performed by Health Canada-certified provider Canvas Lab (Vancouver, BC, Canada) using 8610C GC coupled with a flame ionization detector (GC-FID, SRI Instruments). After calibration, the set of terpenes was used as standards (Restek Cat.#34095). LoQs and LoDs were set at less than 0.01%. Diagrams with retention times are available upon request. Two samples per cannabis cultivar were analyzed.

### 3.5. Cell cultures and Treatments

The THP-1 cells were purchased from American Type Culture Collection (ATCC, Rockville, MD, USA) [25]. This suspension cell line was grown in 100 mm petri dish in the Roswell Park Memorial Institute Medium (RPMI-1640, Cat#350-000-CL, WISENT INC., Quebec, QC, Canada). The complete growth medium was obtained by adding 10% heat-inactivated Premium Grade Fetal Bovine Serum (Cat#97068-085, VWR International LLC, Radnor, PA, USA), according to the ATCC recommendation. The incubation environment for the cells was a humidified atmosphere of 5% CO_2_ at 37 °C. PMA (50 ng/mL, 48 h) was used for the terminal differentiation of monocytes into macrophages. Next, cells were given fresh media for one day. The verification of successful transformation was conducted by evaluating cell adhesion and spreading under the light microscope [26]. 0.5 μg/mL of LPS for 3 h was used for inflammation induction [27]. Cannabis extracts (7 μg/mL) or single cannabinoids (5 μM) were added 20–30 min prior to LPS.

### 3.6. Cell Viability Assay Using Trypan Blue

Trypan blue assay was used for the determination of THP-1 macrophages’ viability and morphology. First, cells were washed with sterile PBS, then trypsinized. The fresh media was added to trypsinized THP-1 macrophages. After that, the cells were centrifuged at 1500 rpm for 5 min at 20 °C. The supernatant was discarded, and the pellet was resuspended in fresh media. The ratio of aliquot cells to trypan blue solution was 1:1. The automated cell counter, LUNA I (Logos Biosystems), was used for counting. The number of viable cells was determined by trypan blue exclusion. The results were shown as a percent of viability.

### 3.7. Multiplex ELISA

Multiplex ELISA was performed for cytokine assessment. Cell culture supernatants were centrifuged at 3000× *g* at 4 °C for 3 min prior to aliquoting to remove debris. The samples were stored at −70 °C. All prepared samples were shipped to Eve Technologies (Calgary, AB, Canada) on dry ice. Human Cytokine Array Proinflammatory Focused 13-plex (HDF13) discovery assay was used for measuring the levels of cytokines upon the first thaw. The results were analyzed using BioPlex 200 [28].

### 3.8. Statistical Analysis

One-way analysis of variance (ANOVA), followed by Tukey post-hoc multiple comparison test, was performed in GraphPad Prism version 6.0 for windows, GraphPad software (La Jolla, CA, USA) [29]. A *p*-value ˂ 0.05 was considered statistically significant.

## 4. Discussion

In this work, we have established an LPS-induced model of the induction of inflammation in THP-1 macrophages. We found that the highest level of induction of pro-inflammatory cytokines was observed at 3 h post-exposure to LPS. We then demonstrated that pre-treatment with CBD, THC, or extracts high in one of these two cannabinoids substantially decreased the induction of cytokines after LPS treatment. CBD was more efficient than THC, and extracts were more efficient than single cannabinoids. IL-6, IL-8, and MCP-1 were most sensitive to pre-treatments with cannabinoids, while IL-1β, IL-10, and TNF-α were much less responsive.

Several other models of inflammation were used in the past, including U937 [30] and Mono Mac 6 [31] cell lines. U937 cells are monocytes derived from the tissue origin of human histiocytic lymphoma. These cells can also be transformed into a macrophage-like phenotype and stimulated with LPS for inflammation induction [32,33]. The response of U937 cells to LPs was not as efficient as in our work; only seven out of thirty-four inflammation-related genes were induced [34]. Stimulation of the Mono Mac 6 cell line, a human monocytic line, with different triggers only upregulated TNF-α, IL-1α/β, and IL-6 [35]. Another type of cell that can be used in inflammation is human peripheral blood mononuclear cell (PBMC)-derived macrophages. These primary cells are considered superior to cell lines as they resemble in vivo settings better. The main disadvantages of primary cells are their short lifespan and heterogenic response among donors, which arise during cell differentiation from progenitors [36]. A comparison between the responses of THP-1 cells and PBMC-derived macrophages to LPS revealed a close correlation in inflammatory gene expression, making a THP-1 cell line a good model for studying LPS-induced changes [34]. 

Many studies have demonstrated the effects of single cannabinoids, such as THC and CBD, on inflammation [37]. Other components of the plant (such as minor cannabinoids, terpenes, terpenoids, flavonoids, and others) may act synergistically with cannabinoids and can be useful from a therapeutic point of view [19]. The modulating effect of these compounds is typically positive, which means that the medicinal effect of the whole plant extract is more significant than the effect of isolated compounds [3,38].

IL-1β is the most prominent pro-inflammatory cytokine and plays a crucial role in inflammasome signaling [39]. IL-6, a pro-inflammatory cytokine, also plays an important role in the inflammatory response of many chronic inflammatory conditions [40]. The TNF-α cytokine stimulates inflammation and is one of the most studied and important pro-inflammatory cytokines [41]. IL-8 and MCP-1 induce chemotaxis of granulocytes to the site of infection and stimulate phagocytosis [42,43]. Our cannabis extracts significantly downregulated the levels of several of these cytokines, with IL-6, IL-8, and MCP-1 being affected the most. The results of this study were similar to those of other studies, where it was shown that cannabinoids and cannabis extracts efficiently inhibit inflammation by suppressing the levels of pro-inflammatory cytokines. For example, cannabis users who suffered from multiple sclerosis had significantly lower levels of many pro-inflammatory cytokines, such as TNF-α, IL-1, IL-6, IL-12, and IFN-γ, and higher levels of the anti-inflammatory cytokine IL-10, in comparison with non-cannabis users [17,44]. Another study reported that cannabis extract high in CBD was superior to pure CBD in reducing the IL-6 and IL-8 levels in an alveolar epithelial cell line, A549, while cannabis extract high in THC used on the same cell line showed only minor anti-inflammatory activity and was more cytotoxic. This study used 5 µg/mL of extracts, while in our study, we used 7 µg/mL [20]. In a 3D EpiDermFT tissue inflammation model, it was shown that extracts high in CBD and THC were effective in reducing the expression of IL-6, IL-1β, MCP-1, and TNF-α [18]. Another study demonstrated that cannabis extract high in CBD was more effective than pure CBD in a zymosan-induced mouse model of inflammation. The extract effectively reduced pain and paw swelling, prevented TNF-α production, and overcame the bell-shaped dose response of CBD [19]. High CBD and high THC extracts were shown to be superior to pure CBD and THC in reducing inflammation in graft versus host disease in an allogeneic bone marrow transplantation model [21].

The data we obtained on IL-1β levels was quite interesting. This was the only cytokine that decreased in response to CBD more efficiently than in response to high CBD extracts or high THC extracts. It remains to be shown whether this is a specific response to our extracts or the nature of the response of this cytokine to cannabinoids. 

It was also curious to note that CBD and THC had no effect on MCP-1 levels, while our extracts significantly downregulated high MCP-1 levels post-LPS. Our data and reports by others demonstrate that MCP-1 can indeed be downregulated by cannabis extracts [45]. MCP-1 was also downregulated by CBD in human epithelial cells (BEAS-2B and NHBE), macrophages (U937), and lung fibroblast cells (HFL-1) exposed to LPS [46]. In contrast, exposure of THP-1 to LPs with subsequent treatment with increasing concentrations of CBD did not reduce MCP-1 levels [47]. Therefore, it appears that the decrease in MCP-1 expression in response to pure cannabinoids may be cell-specific.

Another interesting result was the effect of cannabinoids on IL-10 levels. IL-10 is an anti-inflammatory cytokine that protects cells against profound inflammation. We observed that LPS induced this cytokine, and pure CBD as well as all extracts reduced it; in contrast, THC slightly increased it. Some studies have demonstrated that cannabinoids and cannabis extracts enhance the levels of anti-inflammatory cytokines. For example, in the murine model of bone marrow transplantation, it was demonstrated that pure THC and CBD, as well as cannabis extracts high in CBD and THC, reduced inflammation by reducing IL-17 secretion and enhancing IL-10 secretion [21]. Another study performed on mouse primary bone marrow-derived macrophages stimulated by LPS showed that flavonoids, luteolin and quercetin, increased the levels of IL-10 secretion [48]. It was also shown that THC reduced inflammation in an endotoxemic mouse model by significantly upregulating the plasma level of the anti-inflammatory cytokine IL-10 while suppressing the pro-inflammatory cytokine MCP-1 [49]. Along this line, it was reported that CBD and THC suppressed the secretion of IL-17 but elevated the secretion of IL-10 in a mouse-derived encephalitogenic T cell line [50]. CBD was reported to significantly reduce the plasma levels of pro-inflammatory cytokines (IFN-γ, TNF-α), along with the increase in the levels of anti-inflammatory cytokines (IL-10, IL-4) in diabetic mice [51].

Since LPS was used to induce the inflammatory response, it was expected that it would also induce anti-inflammatory cytokines to counteract high levels of pro-inflammatory cytokines, which are required for cell survival. For effective clearance of pathogens, the pro-inflammatory response is crucial, while an excessive inflammatory response causes tissue damage. Hence, the human body always maintains equilibrium by activating counteractive pathways to bring down the pro-inflammatory response. Since levels of pro-inflammatory cytokines were elevated by LPS in our study, we observed a corresponding increase in IL-10 levels in the LPS group to counteract the increase in pro-inflammatory cytokines. 

We hypothesize that since our extracts and CBD were able to significantly downregulate the levels of several pro-inflammatory cytokines, they reduced the levels of IL-10, possibly to maintain equilibrium. On the other hand, THC was not able to reduce the levels of pro-inflammatory cytokines as effectively as CBD and, hence, did not change the high levels of IL-10 post-LPS. It is also possible that cannabis extracts inhibit all cytokines, regardless of their nature.

## 5. Conclusions and Limitations of This Study

We established the THP-1 macrophage model for the analysis of the anti-inflammatory properties of cannabinoids and cannabis extracts. We found that the studied cannabis extracts significantly prevented the increase in the levels of pro-inflammatory cytokines induced by LPS. It would be important in the future to treat cells with extracts after the induction of inflammation since, in general, there is a need to treat inflammation rather than prevent it. We may also want to compare the effects of extracts with those of well-known anti-inflammatory agents such as dexamethasone. For a better understanding of our results, it would be very beneficial to look at key transcription factors and target genes by performing transcriptome profiling, followed by analyses of relevant protein expression. The most important consideration in the future is to perform this study in vivo and confirm the immunomodulatory activity of the extracts.

## Figures and Tables

**Figure 1 molecules-28-04991-f001:**
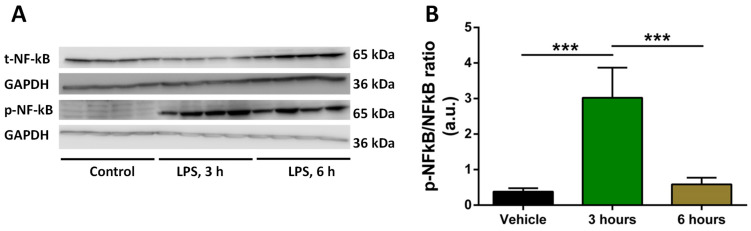
Changes in p-NFκB/NFκB ratio in response to LPS. (**A**) Representative Western blots of p-NFκB and NFκB with GAPDH bands in response to LPS (at 3 h and 6 h) for four independent samples per each treatment are shown. (**B**) Average ratio between p-NFκB and NFκB for control (vehicle), 3 h and 6 h post-treatment with LPS. Data are expressed as mean (n = 4) ± SD. Significance of the differences (***, *p*  <  0.001) was evaluated by one-way ANOVA followed by Tukey post-hoc multiple comparison test.

**Figure 2 molecules-28-04991-f002:**
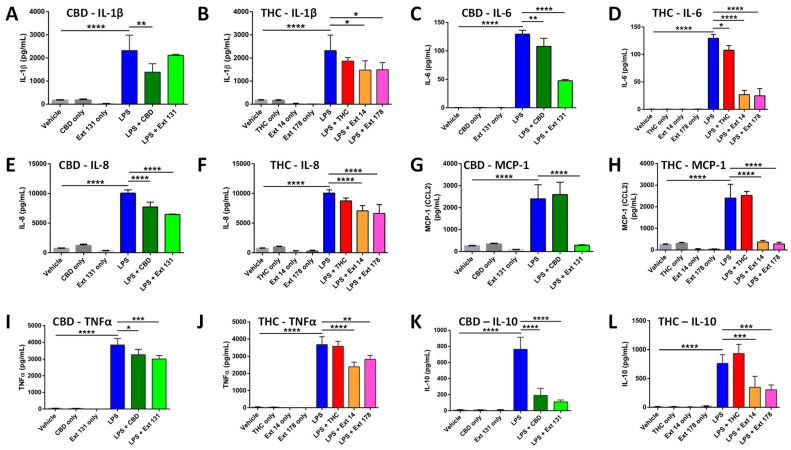
Analysis of cytokine levels using ELISA in response to CBD, THC, or extracts. (**A**)—IL-1β in CBD sample; (**B**)—IL-1β in THC sample; (**C**)—IL-6 in CBD sample; (**D**)—IL-6 in THC sample; (**E**)—IL-8 in CBD sample; (**F**)—IL-8 in THC sample; (**G**)—MCP-1 in CBD sample; (**H**)—MCP-1 in THC sample; (**I**)—TNF-α in CBD sample; (**J**)—TNF-α in THC sample; (**K**)—IL-10 in CBD sample; (**L**)—IL-10 in THC sample. Results are presented as mean of calculated concentration (pg/mL) ± SD of four independent experiments performed in duplicate and quantified by one-way ANOVA followed by Tukey post-hoc multiple comparison test. Significant differences between groups are marked with: * *p*  <  0.05, ** *p*  <  0.01, *** *p*  <  0.001, **** *p*  <  0.0001.

**Table 1 molecules-28-04991-t001:** Concentrations of THC and CBD in extracts of selected cannabis cultivars.

	Total THC, %	Total CBD, %	THC (μM)	CBD (μM)
**#14**	33.35 ± 2.75	2.81 ± 0.23	**7.42** ± **0.56**	0.63 ± 0.04
**#131**	2.11 ± 0.19	19.65 ± 2.05	0.47 ± 0.03	**4.37** **± 0.33**
**#178**	33.98 ± 3.12	1.01 ± 0.09	**7.56** ± **0.64**	0.22 ± 0.02

Concentrations of THC and CBD in the extracts #14, #131, and #178, in % of dry weight, and concentrations of THC and CBD in 7 µg/mL of the extract, in μM.

**Table 2 molecules-28-04991-t002:** Composition of terpenes and their concentrations in each extract as analyzed by GC-FID.

Terpenes in mg/g	#131	#178	#14
α-Pinene	0.295 ± 0.055	0.649 ± 0.12	0.048 ± 0.008
β-Pinene	0.212 ± 0.046	0.245 ± 0.042	0.068 ± 0.012
β-Myrcene	ND	0.361 ± 0.064	0.124 ± 0.042
Limonene	0.262 ± 0.052	0.003 ± 0.001	0.263 ± 0.062
Terpinolene	0.025 ± 0.005	0.008 ± 0.002	0.004 ± 0.001
Linalool	0.058 ± 0.009	0.029 ± 0.005	0.193 ± 0.062
α-Bisabolol	0.003 ± 0.001	0.244 ± 0.06	0.061 ± 0.02
*trans*-Caryophyllene	0.04 ± 0.008	0.076 ± 0.014	0.545 ± 0.08
α-Humulene	ND	0.04 ± 0.009	0.136 ± 0.03
*trans*-Nerolidol	ND	0.008 ± 0.002	0.187 ± 0.04
*cis*-Nerolidol	0.001 ± 0.001	0.003 ± 0.001	ND
Camphene	0.022 ± 0.006	0.025 ± 0.001	0.015 ± 0.005
β-Ocimene	ND	0.089 ± 0.016	ND
Fenchone isomers	ND	0.003 ± 0.001	0.004 ± 0.001
δ-3-Carene	0.506 ± 0.12	0.001 ± 0.001	ND
α-Terpinene	0.006 ± 0.002	0.573 ± 0.11	ND
Eucalyptol	0.002 ± 0.001	ND	ND
γ-Terpinene	ND	0.001 ± 0.001	ND
β-Cymene	ND	0.048 ± 0.009	ND
Camphor isomers	ND	0.118 ± 0.04	ND
Isopulegol	0.01 ± 0.004	0.016 ± 0.004	ND
Caryophyllene oxide	0.017 ± 0.005	0.098 ± 0.003	ND
Valencene	ND	0.01 ± 0.003	ND
Geraniol	0.005 ± 0.002	0.004 ± 0.002	ND
Guaiol	0.029 ± 0.005	0.116 ± 0.04	ND
*trans*-P-Ocimene	0.016 ± 0.004	ND	ND
α-Humulene	0.025 ± 0.006	ND	0.136 ± 0.034
Fenchyl Alcohol	ND	ND	0.036 ± 0.007
Borneol isomers	ND	ND	0.012 ± 0.003
α-Terpineol	ND	ND	0.052 ± 0.009
**Total Terpenes**	**1.534** ± **0.32**	**2.768** ± **0.46**	**1.75** ± **0.25**

ND—not determined.

**Table 3 molecules-28-04991-t003:** Summary of the effects of extracts, CBD, and THC on studied cytokines.

Analyzed Cytokines	CBD	THC	#131	#14	#178
IL-1β	↓	=	=	↓	↓
IL-6	↓	↓	↓	↓	↓
IL-8	↓	=	↓	↓	↓
IL-10	↓	=	↓	↓	↓
MCP-1	=	=	↓	↓	↓
TNF-α	↓	=	↓	↓	↓

↓ shows the decrease, while = shows no significant change.

## Data Availability

All data are available in the main text or the Appendix A.

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
