# Peer review of "The Anti-Inflammatory Effects of Cannabis sativa Extracts on LPS-Induced Cytokines Release in Human Macrophages"

_molecules, 2023, doi:10.3390/molecules28134991_

Round 1
Reviewer 1 Report (Previous Reviewer 1)
I am satisfy with revisions
Author Response
Thank you very much.
Reviewer 2 Report (Previous Reviewer 3)
Dear Author(s)
· I suggest the keywords be according to Mesh terms and rearranged according to the English alphabet.
Dear Author(s)
· The manuscript is exciting but needs minor English editing.
Author Response
We have arranged the keywords according to the alphabetic order and checked that they comply with MeSH. All but tetrahydrocannabinol are as per MeSH. MeSH suggests to use dronabinol instead of tetrahydrocannabinol - we disagree, since dronabinol is a synthetic tetrahydrocannabinol and a drug, while we used HPLC standard tetrahydrocannabinol.
This manuscript is a resubmission of an earlier submission. The following is a list of the peer review reports and author responses from that submission.
Round 1
Reviewer 1 Report
Dear Authors, your work is interesting and properly made, but there are some sections of material and methods that remain obscure. Also, many concepts are repeated similarly in intro and in discussion. Also the MS is plenty of typos. The format of the MS is not that of MDPI. GC approach has some flaws and table 2 is not properly presented.
I suggest to accept the paper after major revisions
Keywords:
to maximize indexing, avoid to use words already present in the title.
L28 cannabis…I am not sure on the appropriate use of vernacular words. Otherwise, it is better to define which species of cannabis.
L28 C. sativa… the first time the species is introduced should come with the full name. Also, Cannabis sativa comes as Cannabis sativa L.
L45-46: Microorganisms express…patterns (PAMPs)…it is better to substantiate the term microorganism with “some bacteria” or “different bacteria”, or “some symbiotic bacteria.
L74; Our work…please revise in “our previous works”
L80: were more sensitive to pre-treatments…what does pretreatments means? Please substantiate.
Table 1: Levels ? it is better to report just concentrations, or relative and absolute concentration.
Table 1. The values must be expressed as mean ± SD.
L94: Concentrations….extract. Is this part a footnotes? Please revise.
Table 2: The title caption should report the methods used. Then how many replicas, if the values are means it should be stated in the footnotes and the value should come as Mean ±SD.
The title header is not proper. Terps is not the full name, thus somewhere must be specified. Or else just report the full name. ND – not determined… Please define the LOD and LOQ for your GCMS approach. Could you add on this table also the retention time.
Figure 1. The blots are just put there, it is not comprehensible to which figure are related or if can stand alone. Figure 1A caption should first indicate the picture at the left and not that at the right.
In caption are mentioned three different level of significant difference, but just one (***) is reported in Figure 1A. Please revise
Figure 1B. What is Ct, please indicate any acronym or code in the figure caption. Why Figure 1B doesn’t have error bar. Please report it.
L126 (n=1) I do not understand the reason not to replicate the test. I think that is better to reserve Figure 1B as supplementary.
L129: samples with expected low….is not comprehensible at all. It is better to write “samples with expected induction of low cytokines levels….
Table 3. I am skeptic on the use of a table as summary of a previous figure. It is better to embed it in Figure 3.
Table 3 title is not fine: “effects of extracts versus CBD/THC on studied cytokines”, but in the table there’s no comparison. It is better to write “effects of extracts, CBD, and THC on studied cytokines ”
L187: ANOVA analysis…did you check for normality and homoscedasticity of variance, before performing ANOVA? Could you include the ANOVA model as supplementary?
L187: Posthoc…post hoc
Materials and methods:
Sigma-Aldrich, Life Technologies, Agilent Technologies, etc…please give details of the suppliers.
L207-208: Cannabis sativa cultivars #14 and #131 were used for the experiments
Please the sole code number I think is not enough to reproduce the experiments. Which strain was used, was this strain deposited. Please give more details.
L214,217, 218, etc: temperature comes as XX °C, not XX°C. Please correct in the whole MS.
L218, 219, etc: please correct commons errors such as 60 mg/ml or 7 ug/ml, the ml should be mL
L227-228: Per each cannabis ….analyzed [18]. This part is not clear, report the n =
L221: High performance liquid chromatography (HPLC). This section needs more clarity. Please give more details. Also check for the many typos, such as dram for gram…
L229: Terpene analysis…please rewrite this part, because like that the method is obscure. Which standard was used? Please give full description of the method. Did you perform SPME? How did you prepare the samples prior analysis? Also, quantifications of compounds with no MS is not possible. GCMS is a quali/quantitative approach, but GC is just qualitative approach.
Please, if an MS was not used, reformulate Table 2, with no absolute values, but peak areas or percentage.
L234 and elsewhere: ((ATCC…please avoid double brackets
L262: p-value…please always report just p
Discussion:
L273:295: I do not understand the reason to report this part. Also because there is no consistency with the paper’s results. If the authors want to introduce this aspect to then state the innovation of the present work, this if fine, but at the end of such narrative block there’s no track of solution to a problem. Please substantiate or remove it or reformulate it.
L296-303: this part is not related to critical comparison and evaluation of results. This part seems a repetition of the concepts of the introduction. Please remove it or reformulate it with your results.
L332-333: The cannabis extracts…a similar field of research. This is not a comparison of result, but of methods. Please remove it or reformulate it.
L334-336: I am also skeptic that this part has to deal with discussion.
L337: This part is unbound to the previous. Please find a sentence to link them
Reviewer 2 Report
This is a resubmission of the article titled 'The anti-inflammatory effects of cannabis sativa extracts on LPS-induced cytokines release in immune cells in vitro'
The article, while interesting conceptually, the lack of clarity in experimental data and significant lack of depth to the research findings continue to dampen the enthusiasm for the manuscript.
Focusing on the responses to prior review of the article here are some considerations.
1. The confusion around Nf-kB blot stems from the issue with data presentation approach adopted by the authors. The font size attributed to labels makes it challenging to read them correctly, and were hence incorrectly understood to be two sets of pNF-kB blots with GAPDH normalization. Further, if pNF-kB and NF-kB are the essential correlates, ideally they would be placed one above the other to provide clarity. It is now clear that this is an data presentation issue which is now resolved. One suggestion is to improve the font clarity.
2. Staying on with the western blots. What is C1-4, 3a-3d and 6a-6d? The authors do not define these annotations in the text of the Figure 1 data or in the figure legend. What is the significance of changes in tNF-kB in the western blots?
3. The GAPDH western blots are not acceptable based on the standards that are typically observed for such a commonly immunoblotted target. For one, there is significant change in the intensity of the signal of GAPDH in both blots (more apparent in top one). The lower GAPDH blot is faded away on the right end. Moreover, with the change in tNF-kB signal in different lanes, it becomes important to now normalize this change in relation to GAPDH. The bands are indiscernible between the individual lanes (compare to pNF-KB blots).
4. This reviewer is still not clear on the purpose of the Figure 1B data. This is typically part of methods optimization that is not presented as a data set for publication consideration. Was IL-6, IL-8, and MCP-1 data not important in Figure 1B for consideration of this multiplex assay? Were they part of the multiplex 13-plex platform? If it is a 13-plex, what are the other cytokines?
5. Figure 2 data is helpful to demonstrate impact of these extracts on cell viability, but more suitable as supplementary data establishing analytical baselines.
Additional minor points
1. In the abstract, the authors make contradicting statements about IL-6. Is it most sensitive or least responsive?
2. Fig S1 is not helpful in conveying any meaningful information. It does not provide the reviewer nor the reader anything of relevance.
Reviewer 3 Report
This paper addresses the anti-inflammatory properties of CBD. The work is interesting, but the mention of the aim(s) of the study can be clarified better. In terms of novelty, the anti-inflammatory of CBD has been reported previously. This work is on human THP-1 macrophages, so the model of study is different, so it can be interesting for the readers of Molecules.
1. Please replace the keywords according to Mesh terms.
2. Please clarify the aim(s) of this work in a more convincing way.
3. There are many words in the text that are not consistent. Please check and align the entire text, figures, and tables for the words that are not the same and also correct some of the words with the wrong typos.
4. The text can be improved to explain the scope and method more clearly. The overall readability of the text can be improved, since there are some problems in the text such as typos and punctuation errors.
5. Generally, the figures are good. A mechanistic figure can help a lot in the height of understanding of the article.
6. The conclusion is consistent with the evidence/arguments. But I would suggest that it should be more focused and concise. The conclusion is currently long.